# Application and renovation evaluation of Dalian's industrial architectural heritage based on AHP and AIGC

Yao Liu[1], Pengjun Wu[1]*, Xiaowen Li[1], Wei Mo[2]

1 School of Plastic Arts, Daegu University, Gyeongsan-si, Gyeongsangbukdo, South Korea, 2 College of art and design, Southwest Forestry University, Kunming, Yunnan, China

* wupengjun@daegu.ac.kr

## Abstract

This paper takes the example of industrial architectural heritage in Dalian to explore design scheme generation methods based on generative artificial intelligence (AIGC). The study compares the design effects of three different tools using the Analytic Hierarchy Process (AHP). It first establishes the key indicator weights for the renovation of industrial architectural heritage, with the criterion layer weights as follows: building renovation 0.230, environmental landscape 0.223, economic benefits 0.190, and socio-cultural value 0.356. Among the goal layer weights, the highest weight is for the improvement of living quality at 0.129, followed by resident satisfaction at 0.096, and educational and display functions at 0.088, while the lowest is for renovation costs at only 0.035. The design schemes are generated using Stable Diffusion, Mid Journey, and Adobe Firefly tools, and evaluated using a weighted scoring method. The results show that Stable Diffusion excels in overall image control, Mid Journey demonstrates strong artistic effects, while Adobe Firefly stands out in generation efficiency and ease of use. In the overall score, Stable Diffusion leads the other two tools with scores of 6.1 and 6.3, respectively. Compared to traditional design processes, these tools significantly shorten the design workflow and cycle, improving design quality and efficiency while also providing rich creative inspiration. Overall, although current generative artificial intelligence tools still have limitations in understanding human emotions and cultural differences, with continuous technological iteration, this method is expected to play a larger role in the design field, offering more innovative solutions for the renovation of industrial architectural heritage.

## Introduction

The traditional architectural design process is complex, requiring designers to independently complete a large amount of work [1]. During the design process, they must consider solutions that meet legal regulations, ecological principles, aesthetic creativity, plant configuration, traffic flow, materials, and costs. In terms of visual representation, the process involves conceptualization, sketching, modeling, and rendering. These tasks demand significant human and

**Data Availability Statement:** All relevant data are within the manuscript.

**Funding:** Yunnan Provincial Science and Technology Department (2021Y318) The funders had no role in study design, data collection and

analysis, decision to publish, or preparation of the manuscript.

**Competing interests:** The authors have declared that no competing interests exist.

material resources [2, 3]. Due to the complexity, even minor adjustments often require designers to repeat certain parts of the design process [4]. Additionally, clients usually ask designers to provide multiple design options for a project, which increases the workload. As a result, this design process and market environment lower the efficiency of design projects, and the overwhelming tasks limit the designer's creativity from being fully expressed [5, 6].

The traditional methods present several key issues that need improvement: 1. Time and efficiency, 2. Repetitive tasks, 3. The demand for multiple design options, and 4. Substantial human and material resource investment. The emergence of Generative AI (AIGC) has replaced many traditional design processes [7], gradually transforming the designer's role from "creator" to "decision-maker." AIGC provides a new approach to design, effectively addressing these issues.

AIGC's characteristics make it particularly useful in assisting with the early, heavy, and repetitive stages of conceptual design and rendering. Specific improvements include: 1. Automated generation of initial concepts: Designers only need to input keywords to quickly generate multiple preliminary designs, shortening the design cycle. 2. Reducing repetitive tasks: AIGC can automate many traditional design steps, such as concept generation, color matching, and structure optimization, allowing designers to adjust only the parameters. 3. Efficient multi-option generation: AIGC enables designers to quickly generate multiple options for clients to choose from, improving client satisfaction. 4. Optimizing resource allocation: AIGC tools typically integrate various AI technologies, such as deep learning and edge detection algorithms, to quickly generate high-quality renderings, reducing reliance on specialized software and manpower. This approach transforms conceptualization directly into visualization, replacing many traditional design processes. By combining AI thinking with the designer's creative thought process, the design cycle is significantly shortened.

To comprehensively evaluate the practical application of generative AI tools, this study has developed a multidimensional evaluation system for the renewal of industrial architectural heritage. This system includes specific indicators such as design quality, generation efficiency, ease of use, cultural and emotional understanding, innovation, and practicality. Using the Analytic Hierarchy Process (AHP) and weighted scoring methods, we conducted a comprehensive evaluation of three generative AI tools to ensure the analysis is thorough and scientific. Through an in-depth empirical analysis of the renewal of industrial architectural heritage in Dalian, the application and improvement suggestions of generative AI tools in real projects are demonstrated.

In summary, the innovative and original contributions of this paper are: 1. Construction of an evaluation system: A multidimensional evaluation system was designed based on the actual needs of industrial heritage renewal, and its effectiveness was verified through practical research. 2. Determination of indicator weights: The Analytic Hierarchy Process was used to scientifically determine the weight of each indicator, providing objective reference points for the application of generative AI in industrial heritage renewal. 3. Empirical analysis: An in-depth empirical analysis was conducted through the renewal of Dalian's industrial architectural heritage, demonstrating the application of generative AI tools in real projects and providing improvement suggestions [4].

## Literature review

The protection and renewal of industrial heritage have gained increasing attention in the field of architecture, as these heritages hold significant historical, cultural, and social value [8]. However, these processes face numerous challenges, such as structural degradation, the need for functional adaptation, and the difficulty of integrating modern requirements while

maintaining historical integrity [9]. Traditional methods are often labor-intensive, time-consuming, and costly. Therefore, there is an urgent need for innovative approaches to simplify these processes and improve protection outcomes, as emphasized by existing research.

Generative AI (AIGC), as a cutting-edge solution, offers tools that automate and enhance various stages of architectural design. AIGC refers to a class of algorithms capable of creating new content based on input data. In the context of architectural design, these algorithms can generate alternative design options, simulate structural impacts, and provide detailed visual effects [10]. Although the application of AIGC in industrial heritage is still a relatively novel field, it has shown significant potential in addressing the pain points of traditional preservation methods.

One of the main advantages of generative AI is its ability to handle complex design tasks traditionally performed by humans. The traditional architectural design process involves multiple stages, including conceptualization, sketching, modeling, and rendering, each requiring substantial human intervention. This is not only time-consuming but also prone to human error. Generative AI tools can automate these tasks, reducing the time and effort needed to produce high-quality designs. Research by Wang et al shows that these tools can significantly shorten design cycles while improving design quality and efficiency [11].

Stable Diffusion, a generative AI tool based on diffusion models, has demonstrated strong control and flexibility in generating architectural designs. Researchers widely agree that this tool excels in producing high-resolution images and handling intricate details. A study by Cao et al on the application of Stable Diffusion in architectural design highlights its ability to significantly reduce the workload for designers when generating complex architectural structure diagrams [12]. Specifically, in the visualization of architectural heritage, Stable Diffusion, through its denoising process, can gradually generate clearer and more detailed renderings. Zhang et al point out that the tool's output can be further optimized through plugins, such as using Control Net's Canny edge detection algorithm to maintain the accuracy of generated images [13]. However, despite its technical excellence, Stable Diffusion has some limitations. According to Hu et al, the tool requires substantial training data and computing power, and without fine-tuning on specific datasets, the generated renderings may deviate from actual design requirements [14].

Based on the features of Stable Diffusion and the project's requirements, the model can be fine-tuned using LORA to improve its performance. Additionally, plugins such as Control Net's Canny (edge detection algorithm) and Depth (depth algorithm) can enhance image details, and the Tiled Diffusion and Tiled VAE plugins can improve image resolution.

MidJourney, driven by Generative Adversarial Networks (GANs), is an AI design tool widely used in the early stages of conceptual creation in architectural design due to its ability to generate highly artistic and visually impactful images [10]. Goodfellow et al laid the theoretical foundation for GANs in their study on 'Generative Adversarial Nets,' explaining the basic architecture and training principles of GANs [15], where the generator learns to create data from random noise, and the discriminator distinguishes real data from generated data. Mahalingaiah et al introduced a progressive training method that significantly improved the quality and resolution of generated images by training GANs at multiple resolution stages [16]. In his later research, Karras proposed the StyleGAN architecture, which better controls the features of generated images, such as textures and colors [17].

In practice, Caires et al suggest that combining design thinking methods with AIGC enables users to generate innovative and artistically expressive design concepts during the ideation phase, enhancing the design experience [18]. Jiang et al note that MidJourney's adversarial generation mechanism can quickly produce diverse design options [19], providing architects with a wealth of inspiration. Petráková also mention that the tool demonstrates excellent

composition and color coordination abilities in generating conceptual design drawings, making it highly advantageous in the early stages of architectural design [20].

Adobe Firefly, a generative AI tool integrated into Adobe Creative Cloud, is mainly used for quickly generating prototype designs and conceptual plans. Through deep learning and edge detection algorithms, it can generate efficient design options in a short amount of time and is known for its ease of use. A quantitative analysis by Poredi shows that Adobe Firefly is a highly efficient AIGC tool that is easy to learn and interact with, making it ideal for generating multiple design options in the early stages of design, thereby significantly reducing design time [21]. Research by Meron and Song indicates that this tool can be used as a standalone web application or directly within applications such as Photoshop and Illustrator, seamlessly integrating Firefly's generative features into familiar working environments [22, 23]. This integration allows users to harness AI technology to enhance the creative process, improving productivity. Haze et al. believe that Adobe Firefly's ease of use and professional capabilities help enhance users' learning and creativity [24]. Moreover, Firefly is integrated into the Adobe Express platform, providing users with a suite of user-friendly tools for creating professional-quality visual content.

Despite its significant advantages, generative AI still faces certain limitations and challenges. One of the main challenges is incorporating cultural and emotional elements into AI-generated designs [9]. Current AI tools often lack a comprehensive understanding of the cultural and emotional significance of heritage sites, which may result in aesthetically pleasing designs that are culturally inappropriate. Researchers such as Li et al are working to integrate cultural heritage data into AI algorithms to enhance their cultural and emotional sensitivity [25]. This includes training AI models on datasets containing cultural and historical information to generate designs that are both visually appealing and culturally relevant. Another challenge is that applying generative AI in industrial heritage projects requires interdisciplinary collaboration. Preservation and renewal projects typically involve multiple stakeholders, including architects, historians, engineers, and community members. Effective communication and collaboration are crucial to the success of such projects. Generative AI tools facilitate this collaboration by providing detailed visualizations and simulations, promoting the sharing and discussion of ideas among stakeholders, ensuring that all viewpoints are considered, and the final design meets the needs and expectations of all parties [19].

In addition, data quality and availability are issues that need to be addressed [26]. Generative AI models require large amounts of high-quality data to function effectively, including data on existing structures, materials, and historical records [27]. In many cases, this data may be incomplete or unavailable, limiting the effectiveness of AI tools [24]. Researchers are exploring advanced data collection techniques and developing more complex AI models capable of handling incomplete or imperfect data to address this issue.

Overall, generative AI offers significant potential benefits for the preservation and renewal of industrial architectural heritage. By automating many tasks traditionally performed by human designers, these tools can reduce the time and cost of preservation projects, improve design quality and precision, and provide abundant creative inspiration. Additionally, the ability of generative AI to quickly and efficiently generate multiple design alternatives helps ensure the best solution is found for each project.

## Materials and methods

### Model training and plan collection

**Experimental environment.** Unlike online image generation tools such as Stable Diffusion, MidJourney, and Adobe Firefly, Stable Diffusion requires a local computer with a certain

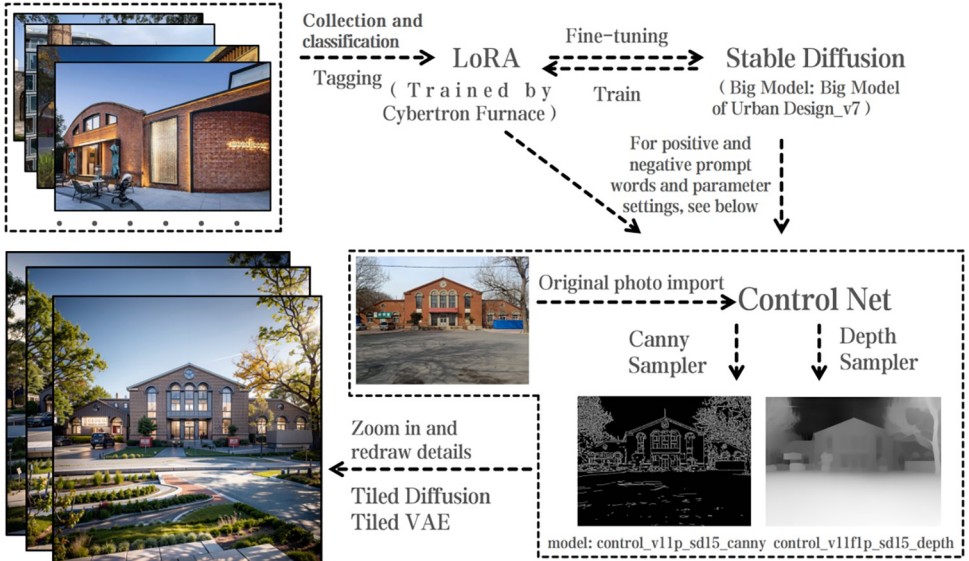

**Fig 1. Stable diffusion drawing process.**

level of computational power [28]. Therefore, the chosen setup includes a CPU: Intel Core i7-14700K, GPU: NVIDIA GeForce RTX 4070 Super 12GB, and the programming language Python, version 3.10.8, as the environment for image generation.

**Generative AI design practice.** The image generation using Stable Diffusion primarily involves three steps. First, collect relevant and high-quality cases and train a fine-tuned model. Second, convert the design requirements into prompts that generative AI can understand or activate relevant plugins to meet the image generation requirements [29]. Finally, utilize the highly efficient image generation method to produce a large number of design plans, select the optimal plan, and then upscale the resolution and refine the details as needed (Fig 1).

Stable Diffusion's open-source nature has attracted numerous developers who have created a wide range of plugins, making text-to-image and image-to-image generation highly controllable. For designers, one standout plugin is Control Net, developed by a research team at Stanford University. This plugin integrates various samplers, and for this particular research, two samplers from Control Net—Canny (for hard edges) and Depth—are utilized. In addition, to meet the output quality requirements, Tiled Diffusion and Tiled VAE plugins are used. These plugins help redraw and upscale images, improving resolution while enhancing overall quality and detail structure.

By configuring these settings and entering the necessary prompts, such as: industrial architecture, masterpiece, best quality, trees, succulents, grass, flowers, rocks, HDR-based rendering, UHD, 8K, best quality, outdoor lawn, (cherry blossom:1.3), blue sky, nature, cloudy sky, highly detailed, studio lighting, clear focal points, high-end, a local computer can generate an image in about ten seconds. This setup optimizes workflows and improves design efficiency. Batch generation of design schemes reduces the difficulty of creative design, accelerating decision-making.

When using MidJourney for image generation, the quality and relevance of the generated images depend on several factors, including the choice of reference images, prompt inputs, and selection of the best from batch-generated images. First, based on the requirements of each scene, it is crucial to choose reference images similar to the target scene. The following aspects should be considered when selecting reference images:

1. Style Matching: The reference image should closely match the overall style of the target scene.

2. Detail Features: Details in the reference image, such as the type of buildings, environmental elements (e.g., roads, vegetation), and lighting effects, should align with the requirements of the target image.

3. Resolution and Clarity: High-resolution and clear reference images provide richer detail, enhancing the quality of the final generated image.

Second, prompt inputs should specify requirements for the main subject, foreground elements, overall style, resolution, composition details, perspective, and lighting effects. For example, in the first scene, a prompt could be: the main body of the picture is a two-story industrial building, with roads, landscapes and flowers in the foreground, reasonable landscape design, photo-realistic style, 8K, architectural visualization/Architectural rendering, high detail, super wide angle, mid shot, beautiful lighting,—ar 4:3—v 6—quality 2.

Finally, by generating multiple images, you can select the ones that best meet the expected outcomes. If the selected images still fall short, adjustments to the prompts, reference images, or generation parameters can be made for further refinement until a satisfactory result is achieved.

During the training process, since Stable Diffusion is based on the regeneration of existing models, there are relatively few models related to industrial buildings available online. Therefore, I used the LoRA training software "Cybertron Furnace" to preprocess and train the industrial heritage building model.

The first step involved data collection, where I collected over 400 images related to industrial heritage buildings from the internet. To meet the requirements for using the dataset, I invited relevant professional designers, master's students, and doctoral students to classify the image types and eliminate duplicate and low-quality images.

The second step was preprocessing the collected images of industrial heritage buildings using Photoshop to remove unrelated information such as watermarks, logos, and people.

In the third step, I adjusted the base model to the larger model "Stable Diffusion Urban Design Model_V7," naming the prompt "Industrial Heritage Building Update," and selected "Architecture" for the automatic XYZ and sample preview.

The fourth step involved selecting a resolution of "512×512" in the software, choosing the "Focus Crop" mode, setting TGA to "Automatic TGA," and clicking to start training.

In the fifth step, through multiple batches of training and parameter adjustments, I selected the most suitable LoRA model for subsequent image generation. The comparison of the LoRA training results is shown in Fig 2.

The training results indicate that the overall performance of the LoRA models met the expected goals, although variations were observed across specific scenes [30]. Models "Industrial Heritage Building Update—1" and "Industrial Heritage Building Update—2" encountered issues such as proportion imbalance, resulting in the generation of irrelevant buildings and water surfaces. Model "Industrial Heritage Building Update—3" performed adequately overall but exhibited some roughness in detail, particularly in Scene 2, where the integration of buildings and environments lacked depth. In contrast, models "Industrial Heritage Building Update—4" and "Industrial Heritage Building Update—5" showcased a more natural relationship between buildings and their surroundings, with improved light and shadow clarity, detail layers, and depth. Among these, "Industrial Heritage Building Update—5" excelled in image richness and stability, making it the preferred choice for subsequent image generation.

Thanks to the open-source nature of Stable Diffusion, numerous plugins have emerged that enable controllable functions such as text-to-image and image-to-image generation [31].

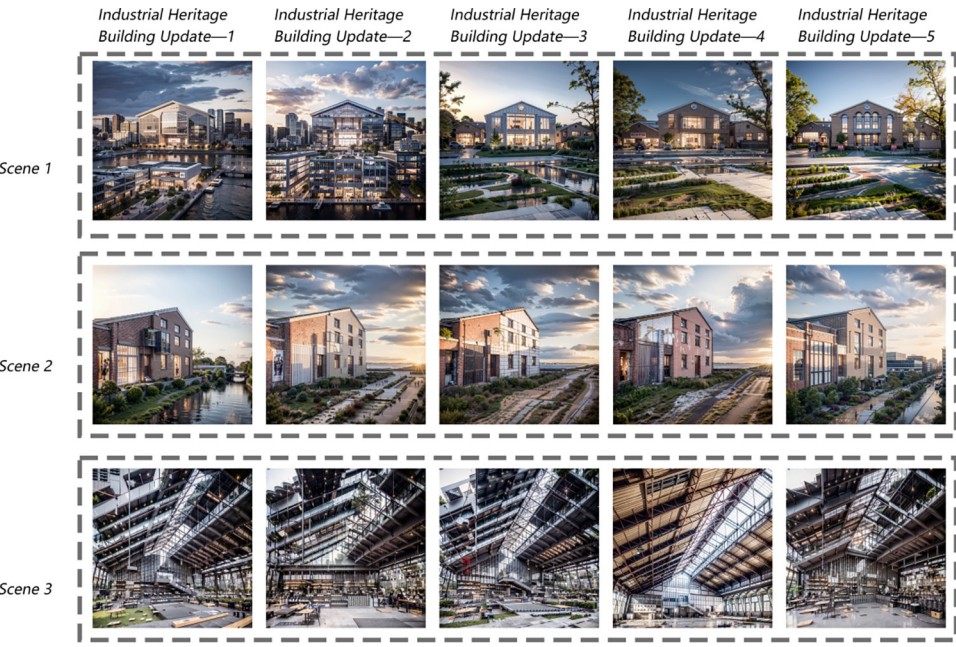

**Fig 2. Analysis and evaluation index system of interior design renderings.**

Notably, the Control Net plugin, developed by a research team at Stanford University, incorporates multiple samplers. For this project, two samplers from Control Net—Canny (for hard edges) and Depth (for depth)—were employed. Additionally, the Tiled Diffusion and Tiled VAE plugins were used to meet image output requirements, allowing for the redrawing and upscaling of output files, which improved resolution and detail.

By configuring these tools and inputting specific prompts—such as industrial architecture, nature elements, and high-quality specifications—images can be generated approximately every ten seconds, optimizing workflow and design efficiency [32]. This batch generation method facilitates the creative process by producing multiple design proposals quickly, thereby expediting decision-making.

In generating images with MidJourney, several factors influence quality and alignment, including reference image selection, prompt input, and image selection from generated batches.

First, selecting reference images that closely match the target scene's style and detail features is essential. The reference images should include relevant elements like building types, environmental features, and light effects, with high resolution to ensure richer detail.

Second, prompts must clearly outline image subjects, foreground elements, style, resolution, composition, perspective, and lighting effects. For instance, a prompt for the first scene might specify a two-story industrial building with surrounding landscapes and flowers, emphasizing photorealism and detailed architectural rendering. By generating multiple versions, users can select the best images and refine prompts or parameters as needed until satisfactory results are achieved.

Adobe Firefly offers a straightforward image generation method, utilizing functionalities similar to the Canny edge detection algorithm in the Control Net plugin, ensuring structural accuracy in generated images [33]. Its user-friendly design allows prompt input in Chinese, significantly lowering the entry barrier for users unfamiliar with complex syntax. This accessibility enables even novices to produce images that meet their needs.

Firefly consolidates complex functions into a simple button interface, allowing users to generate images by clicking relevant buttons rather than manually entering detailed parameters. For example, to generate an image for the first scene, a user can upload a reference photo, click the necessary buttons for style and angle, and input prompts like "industrial architecture, outdoor, landscape, flowers, roads, sky." This streamlined approach simplifies the image generation process and enhances the overall user experience.

**Collection and organization of proposals.** To better determine the future research direction for the update of industrial heritage buildings, we will collect opinions and suggestions through interviews, in addition to reviewing existing literature [11, 34]. This approach will not only help us understand the shortcomings and areas for improvement in current research but also provide valuable insights into future research directions. The interviewees will cover multiple fields, including experts in heritage conservation, urban planners, historians, community residents, and tourists. The interviews will be conducted face-to-face, with a total of 74 relevant individuals providing their opinions and suggestions. Based on the objectives and targets of the interviews, we designed the following core questions:

1. What do you believe are the main achievements in the update of industrial heritage buildings?

2. What challenges and issues have you encountered during the design or usage process?

3. In which areas do you think the existing research is lacking?

4. What suggestions and expectations do you have for future research directions?

5. How do you think public awareness and participation in industrial heritage buildings can be improved?

By refining and categorizing the expressed information from the interviewees, we collected the following user keywords:

1. Historical and cultural value: historical heritage, cultural identity, cultural preservation, historical atmosphere;

2. Functionality and practicality: space utilization, user convenience, safety assurance;

3. Architectural aesthetics: exterior design, visual effects, environmental harmony;

4. Economic and social benefits: economic feasibility, social impact, enhancement of public services;

5. Environmental protection and sustainability: sustainable development, environmental protection;

6. User experience: smooth flow of pedestrian traffic, service quality and attitude;

7. Community activity space: educational function, tourism potential, government support.

Based on the interview results and in conjunction with generative artificial intelligence methods, eight master's and doctoral students in the team generated several proposals. These proposals utilized different generative AI tools, and each student adhered strictly to the predetermined scene requirements during the generation process, repeatedly adjusting the input prompts and parameters to achieve the best results across multiple batches of generation. Ultimately, two proposals were selected from each generative AI tool for subsequent evaluation and discussion using the Analytic Hierarchy Process (Fig 3).

## Evaluation system construction

**Selection of evaluation indicators.**　The selection of evaluation indicators is derived from three sources: the aforementioned interview results, policy documents issued by various levels of government, and a collection of highly cited literature and relevant books from both domestic and international sources [35]. Policy documents include the "National Industrial Heritage Management Measures" issued by the Ministry of Industry and Information Technology in March 2023, the "Implementation Plan for Promoting the Protection and Utilization of Industrial Heritage in Old Industrial Cities" formulated by the National Development and Reform Commission in June 2020, and the "Interim Measures for the Management of Industrial Heritage in Dalian" issued by the Dalian Municipal Government in 2020 [36].

Relevant domestic literature encompasses Wu Qian's "Post-Use Satisfaction Evaluation of Old Industrial Plant Regeneration" [32]. Jiang Nan and Wang Jianguo's " Comprehensive Evaluation of Modern Architectural Heritage Protection and Reuse [37]."

Internationally, pertinent literature includes Claver's "Multicriteria Decision Tool for Sustainable Reuse of Industrial Heritage into Its Urban and Social Environment," Mushtaha's "Application of the Analytic Hierarchy Process to Developing Sustainability Criteria and Assessing Heritage and Modern Buildings in the UAE," and Stanojevic's "Developing a Multi-Criteria Model for the Protection of Built Heritage from the Aspect of Energy Retrofitting."

Through a review of these policies and literature, conducting field research, and consulting the opinions of relevant experts and scholars, a hierarchical analysis of industrial heritage building updates was conducted, leading to the deduplication, extraction, and classification of evaluation indicators [5, 25]. This resulted in the establishment of specific indicators for the update of industrial heritage in Dalian, as shown in Table 1.

Using the Analytic Hierarchy Process (AHP) for analysis, the structural model is composed of three layers: the goal layer, the criterion layer, and the indicator layer. The goal layer represents the highest layer (first-level indicators), the criterion layer is the middle layer (second-level indicators), and the indicator layer is the lowest layer (third-level indicators). The factors at each level are interrelated, collectively forming the evaluation model for the renewal of industrial heritage in Dalian. Descriptions and specific sources of the layered indicators are shown in Table 2.

**Calculation methods and consistency test of indicators.**　Firstly, there are three methods for calculating indicator weights in the Analytic Hierarchy Process (AHP): arithmetic mean method (sum-product method), geometric mean method (root square method), and eigenvalue method. In this analysis, the arithmetic mean method was selected to determine weights. This method is conducted in three steps, as follows [34]:

1. Normalize the indices of each column of the matrix:

$$\bar{C}ij = \frac{Cij}{\sum_{i=1}^{n} Cij}(ij = 1, 2 \ldots \ldots n) \tag{1}$$

2. Sum up the indices of each row of the matrix:

$$\bar{W} = \sum^{n} Cij(i = 1, 2 \ldots \ldots n) \tag{2}$$

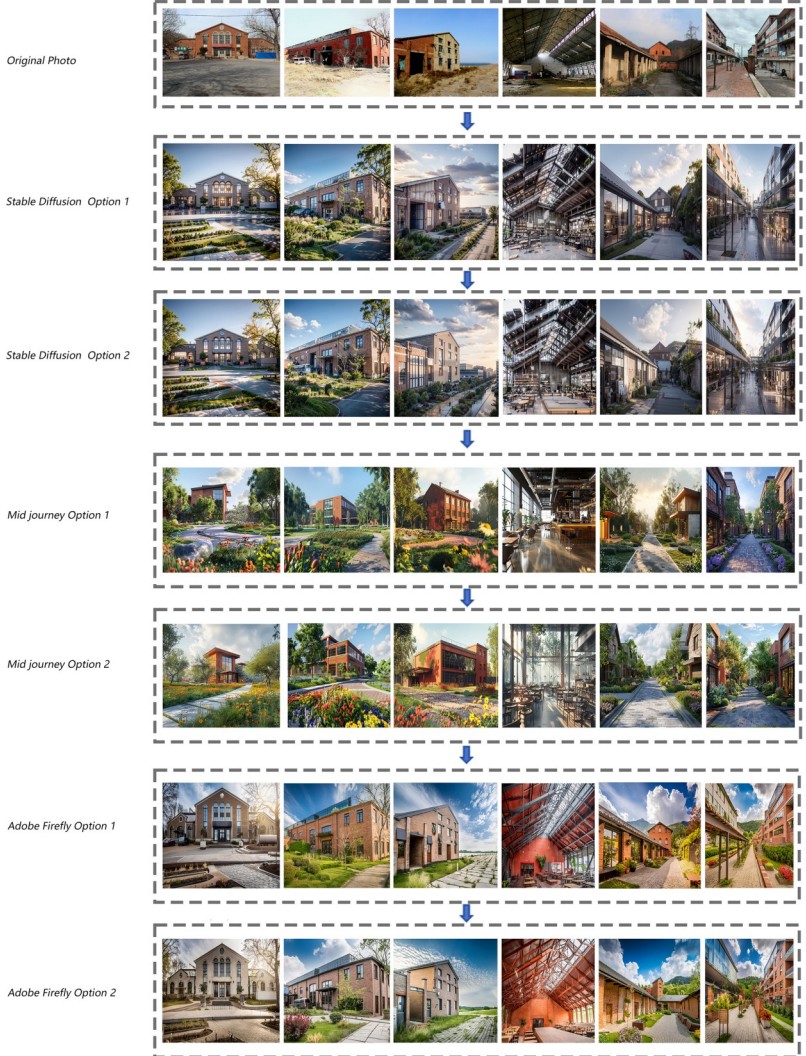

**Fig 3. Analysis and evaluation index system of interior design renderings.**

3. Calculate the weight values:

$$W_i = \frac{\bar{W}_i}{\sum^n \bar{W}_j}(i = 1, 2 \ldots \ldots . n) \tag{3}$$

4. Next, calculate the largest eigenvalue of the matrix using the following method:

$$\lambda_{max} = \sum_1^n \frac{(AW)_i}{nW_i} \tag{4}$$

5. Finally, to ensure the consistency of the judgment matrix, it is necessary to perform a consistency check on the eigenvalues. When CI (Consistency Index) is 0, there is complete consistency. When CI is close to 0, there is satisfactory consistency. A higher CI indicates

greater inconsistency.

$$CI = \frac{\lambda_{max} - n}{n - 1} \tag{5}$$

Since all indices in this matrix are 4, according to Table 3, the random consistency index (RI) value is found to be 0.90. Using the formula CR = (CI / RI), if CR < 0.1, it indicates that the calculated weight values are consistent and pass the consistency check.

**Calculation of weights for each criterion.** To determine the relative importance of each criterion, pairwise comparison matrices were established for each hierarchical level. Each criterion was compared pairwise using a 9-point scale to measure its importance, as shown in Table 4. Ten experts participated in the scoring process. Detailed scores from the first expert are provided in Tables 5 to 9 below.

As shown in Table 5, $\lambda_{max}$ of criterion level (B) is 4.033. The actual consistency index (CI) is 0.010, and the consistency ratio (CR) is 0.012, which is less than 0.10. This passes the consistency check.

As shown in Table 6, the $\lambda_{max}$ for Building Renovation (B1) is 4.017. The actual consistency index (CI) is 0.006, and the consistency ratio (CR) is 0.006, which is less than 0.10. This passes the consistency check.

As shown in Table 7, for environmental aesthetics (B2), the $\lambda_{max}$ is 4.052. The actual consistency index (CI) is 0.018, and the consistency ratio (CR) is 0.019, which is less than 0.10. This passes the consistency check.

As shown in Table 8, for economic benefits (B3), the $\lambda_{max}$ is 4.060. The actual consistency index (CI) is 0.020, and the consistency ratio (CR) is 0.022, which is less than 0.10. This passes the consistency check.

As shown in Table 9, for social and humanistic values (B4), the $\lambda_{max}$ is 4.083. The actual consistency index (CI) is 0.028, and the consistency ratio (CR) is 0.031, which is less than 0.10. This passes the consistency check.

Following the same method, collect the completed questionnaires from the ten experts and calculate the weight values. After passing all consistency checks, compute the average of each

**Table 1. Hierarchical breakdown of architectural renewal needs for Dalian 523 factory.**

| First-level Needs (A) | Second-level Needs (B) | Third-level Needs (C) |
|---|---|---|
| Comprehensive Evaluation of Industrial Heritage Renewal (A) | Building Renovation (B1) | Architectural Continuity (C1) |
| | | Functional Facilities (C2) |
| | | Ventilation and Lighting (C3) |
| | | Insulation (C4) |
| | Environmental Character (B2) | Interaction with Surrounding Landscape (C5) |
| | | Space Layout Integrity (C6) |
| | | Environmental Greening (C7) |
| | | Circulation Flow (C8) |
| | Economic Benefits (B3) | Renovation Cost (C9) |
| | | Maintenance Cost (C10) |
| | | Development Potential (C11) |
| | | Regional Economy (C12) |
| | Social and Cultural Value (B4) | Resident Satisfaction (C13) |
| | | Educational and Exhibition Functions (C14) |
| | | Openness of Space (C15) |
| | | Improvement of Living Quality (C16) |

**Table 2. Description and source of architectural heritage renewal indicators for523 factory.**

| Third Level Requirement (C) | Indicator Description | Source of Indicator |
|---|---|---|
| Architectural Continuity (C1) | Whether the historical characteristics of the original building are respected, preserving its representative appearance and style to ensure the continuity of historical and cultural value. | "Interim Measures for the Management of National Industrial Heritage" |
| Functional Facilities (C2) | New functions should adapt to the original building's structure and layout, enrich urban connotations, highlight urban characteristics, and transform from "industrial rust belt" to "livelihood show belt". | "Implementation Plan for Promoting the Protection and Utilization of Industrial Heritage in Old Industrial Cities" |
| Ventilation and Lighting (C3) | Optimization of ventilation and lighting in industrial buildings to enhance indoor comfort. Includes the addition of new windows, skylights, or ventilation devices to improve air circulation and natural lighting. | "Satisfaction Evaluation after Reuse of Old Industrial Factory Areas" |
| Insulation (C4) | Updating thermal insulation systems in industrial buildings to improve energy efficiency. Utilization of modern materials and technologies such as double-glazed windows, insulation materials, and efficient HVAC systems. | "Developing multi-criteria model for the protection of built heritage from the aspect of energy retrofitting" |
| Interaction with Surrounding Landscape (C5) | Strive for harmony with natural and artificial surroundings. Design, color, and materials should blend harmoniously with the surrounding environment. | "Developing a Landscape Sustainability Assessment Model Using an Analytic Hierarchy Process in Korea" |
| Space Layout Integrity (C6) | Preservation of the original spatial layout without compromising its structure and functionality. Modifications should maintain overall integrity. | "Evaluation Study of Renovation Plans for Old Industrial Buildings with Functional Replacement" |
| Environmental Greening (C7) | Selection and reasonable arrangement of plant species such as gardens, lawns, and trees according to environmental conditions. Aims to protect and improve the ecological environment. | "Multicriteria decision tool for sustainable reuse of industrial heritage into its urban and social environment" |
| Circulation Flow (C8) | Flow paths for personnel and traffic to ensure convenience and safety. | "Comprehensive Evaluation of Protection and Reuse of Modern Architectural Heritage" |
| Renovation Cost (C9) | Includes material costs (e.g., materials, labor), site scale, historical value, structural condition, technology, and engineering. | "Comprehensive Evaluation of Protection and Reuse of Modern Architectural Heritage" |
| Maintenance Cost (C10) | Costs required for regular maintenance and upkeep after site renewal. Includes cleaning, repairs, painting, and maintenance of equipment. | "The research on regional conservation planning of urban historical and cultural areas based on GIS" |
| Development Potential (C11) | Potential economic, social, and cultural benefits. Includes attracting investment, creating job opportunities, promoting tourism, and enhancing urban image. | "Application of the analytic hierarchy process to developing sustainability criteria and assessing heritage and modern buildings in the UAE" |
| Regional Economy (C12) | Whether it can drive surrounding industries, enhance regional attractiveness, and increase tax revenue. | "Interim Measures for the Management of Industrial Heritage in Dalian" |
| Resident Satisfaction (C13) | Satisfaction of original residents with the industrial site renewal plan. Opportunities for feedback and participation in decision-making to ensure the plan meets resident needs. | "Evaluation Study of Replacement Plans for Old Industrial Building Transformation" |
| Educational and Exhibition Functions (C14) | Consideration of industrial sites as carriers of historical and cultural education. Includes the need for museums, exhibition halls, or other educational facilities. | "Interim Measures for the Management of National Industrial Heritage" |
| Openness of Space (C15) | Whether the renovated industrial site is open to the public. Includes public squares, parks, or other open spaces. | "Research on Cultural Industry" |
| Improvement of Living Quality (C16) | Whether it has improved the living standards and hygiene of surrounding residents. | "Research on Evaluation System for National Health Cities" |

criterion's weight values as the final weights of the elements, as shown in Table 10. Finally, based on these weights, determine the weights for generating AI prompts, thereby influencing the final design proposal [11].

**Table 3. Random consistency index value.**

| Matrix Order (n) | 3 | 4 | 5 | 6 | 7 | 8 | 9 |
|---|---|---|---|---|---|---|---|
| Random Consistency Index (RI) | 0.58 | 0.90 | 1.12 | 1.24 | 1.32 | 1.41 | 1.45 |

**Table 4. The scale and meaning of analytic hierarchy process.**

| Scale | Definition and Explanation |
|---|---|
| 1 | Both criteria are equally important. |
| 3 | Vertical criterion is slightly more important than horizontal criterion. |
| 5 | Vertical criterion is noticeably more important than horizontal criterion. |
| 7 | Vertical criterion is significantly more important than horizontal criterion. |
| 9 | Vertical criterion is extremely more important than horizontal criterion. |
| Intermediate values (2, 4, 6, 8) | Scales between the above judgments. |
| Reciprocal values (1/2, 1/3,…) | Opposite scenario where horizontal criterion is more important than vertical criterion in the above importance levels. |

**Table 5. Weight values of each indicator in the criterion layer.**

| Criterion level | Building Renovation (B1) | Environmental Character (B2) | Economic Benefits (B3) | Social and Cultural Value (B4) | Weight values |
|---|---|---|---|---|---|
| Building Renovation (B1) | 1 | 5/7 | 5/3 | 5/7 | 0.234 |
| Environmental Character (B2) | 7/5 | 1 | 6/4 | 3/4 | 0.272 |
| Economic Benefits (B3) | 3/5 | 4/6 | 1 | 4/7 | 0.168 |
| Social and Cultural Value (B4) | 7/5 | 4/3 | 7/4 | 1 | 0.326 |

**Table 6. The weight values of various sub evaluation indicators in building renovation.**

| Building Renovation (B1) | Architectural Continuity (C1) | Functional Facilities (C2) | Ventilation and Lighting (C3) | Insulation (C4) | Weight values |
|---|---|---|---|---|---|
| **Architectural Continuity (C1)** | 1 | 7/4 | 6/5 | 6/5 | 0.306 |
| **Functional Facilities (C2)** | 4/7 | 1 | 3/6 | 3/4 | 0.164 |
| **Ventilation and Lighting (C3)** | 5/6 | 2 | 1 | 6/4 | 0.305 |
| **Insulation (C4)** | 5/6 | 4/3 | 4/6 | 1 | 0.225 |

**Table 7. The weight values of each sub evaluation indicator in the environmental landscape.**

| Environmental Character (B2) | Interaction with Surrounding Landscape (C5) | Space Layout Integrity (C6) | Environmental Greening (C7) | Circulation Flow (C8) | Weight values |
|---|---|---|---|---|---|
| **Interaction with Surrounding Landscape (C5)** | 1 | 4/3 | 5/3 | 5/7 | 0.276 |
| **Space Layout Integrity (C6)** | 3/4 | 1 | 6/4 | 4/5 | 0.238 |
| **Environmental Greening (C7)** | 3/5 | 4/6 | 1 | 5/6 | 0.188 |
| **Circulation Flow (C8)** | 7/5 | 5/4 | 6/5 | 1 | 0.297 |

## Results and discussion

The above plans will be scored using a weighted scoring method [38, 39]. First, a questionnaire will be distributed to ten experts to rate each of the sixteen indicators for each plan, with a maximum score of nine for each indicator. The obtained scores will then be multiplied by the corresponding "layer weight" in Table 10 to calculate the weighted score. Finally, the scores for the four indicators will be multiplied by their corresponding "weights" and summed to derive the criterion layer scores and total scores for each plan. The specific results are shown in Table 11.

**Table 8. The weight values of each sub evaluation indicator in economic benefits.**

| Economic Benefits (B3) | Renovation Cost (C9) | Maintenance Cost (C10) | Development Potential (C11) | Regional Economy (C12) | Weight values |
|---|---|---|---|---|---|
| **Renovation Cost (C9)** | 1 | 3/5 | 3/7 | 4/8 | 0.145 |
| **Maintenance Cost (C10)** | 5/3 | 1 | 6/4 | 6/7 | 0.287 |
| **Development Potential (C11)** | 7/3 | 2/3 | 1 | 5/8 | 0.239 |
| **Regional Economy (C12)** | 2 | 7/6 | 8/5 | 1 | 0.329 |

**Table 9. The weight values of various sub evaluation indicators in social and humanistic values.**

| Social and Cultural Value (B4) | Resident Satisfaction (C13) | Educational and Exhibition Functions (C14) | Openness of Space (C15) | Improvement of Living Quality (C16) | Weight values |
|---|---|---|---|---|---|
| **Resident Satisfaction (C13)** | 1 | 6/8 | 6/4 | 4/7 | 0.218 |
| **Educational and Exhibition Functions (C14)** | 8/6 | 1 | 7/5 | 8/5 | 0.320 |
| **Openness of Space (C15)** | 2/3 | 5/7 | 1 | 4/7 | 0.175 |
| **Improvement of Living Quality (C16)** | 7/4 | 5/8 | 7/4 | 1 | 0.287 |

**Table 10. Comprehensive weight and ranking of Dalian 523 factory's renewal.**

| First-level Needs (A) | Second-level Needs (B) | Weight values | Third-level Needs (C) | Weight at this level | Comprehensive weight | Ranking |
|---|---|---|---|---|---|---|
| Comprehensive Evaluation of Industrial Heritage Renewal (A) | Building Renovation (B1) | 0.230 | Architectural Continuity (C1) | 0.331 | 0.077 | 4 |
| | | | Functional Facilities (C2) | 0.238 | 0.056 | 9 |
| | | | Ventilation and Lighting (C3) | 0.239 | 0.056 | 8 |
| | | | Insulation (C4) | 0.192 | 0.045 | 15 |
| | Environmental Character (B2) | 0.223 | Interaction with Surrounding Landscape (C5) | 0.282 | 0.062 | 5 |
| | | | Space Layout Integrity (C6) | 0.242 | 0.053 | 11 |
| | | | Environmental Greening (C7) | 0.227 | 0.050 | 13 |
| | | | Circulation Flow (C8) | 0.249 | 0.055 | 10 |
| | Economic Benefits (B3) | 0.190 | Renovation Cost (C9) | 0.219 | 0.044 | 16 |
| | | | Maintenance Cost (C10) | 0.224 | 0.045 | 14 |
| | | | Development Potential (C11) | 0.254 | 0.052 | 12 |
| | | | Regional Economy (C12) | 0.304 | 0.062 | 6 |
| | Social and Cultural Value (B4) | 0.356 | Resident Satisfaction (C13) | 0.244 | 0.084 | 2 |
| | | | Educational and Exhibition Functions (C14) | 0.243 | 0.083 | 3 |
| | | | Openness of Space (C15) | 0.168 | 0.058 | 7 |
| | | | Improvement of Living Quality (C16) | 0.345 | 0.118 | 1 |

In terms of "architectural renovation," Stable Diffusion significantly outperforms the other two tools, particularly excelling in overall image control. Mid Journey struggles to effectively reference the structure of the original photo, resulting in lower scores for the layered indicator of "continuity of architectural style," leads to a substantial difference in overall scores. For "environmental character," all three tools achieve a relatively harmonious environment, showcasing unique styles in circulation design and landscape integration, with minimal differences among them.

**Table 11. Score of three generative artificial intelligence tool works.**

| Tool Name | Solution Name | Building Renovation | Environmental Character | Economic Benefits | Social and Cultural Value | Total Score |
|---|---|---|---|---|---|---|
| **Stable Diffusion** | 1 | 6.213 | 6.076 | 5.544 | 6.558 | 6.178 |
| | 2 | 6.380 | 5.763 | 5.751 | 6.921 | 6.316 |
| **Mid journey** | 1 | 5.072 | 5.154 | 5.342 | 4.356 | 5.174 |
| | 2 | 4.202 | 5.126 | 4.846 | 4.959 | 5.023 |
| **Adobe Firefly** | 1 | 5.984 | 5.675 | 5.129 | 4.851 | 5.349 |
| | 2 | 5.462 | 5.425 | 4.969 | 5.230 | 5.277 |

Regarding "economic benefits," none of the tools effectively balance the layered indicators (renovation cost, maintenance cost, development potential, regional economy) effectively. Stable Diffusion scores lower in the first two indicators but higher in the latter two, while Adobe Firefly exhibits the opposite trend. Mid Journey holds a moderate position. This discrepancy arises because Stable Diffusion makes excessive alterations to images, while Adobe Firefly retains the original appearance as much as possible.

For "social and cultural value," Stable Diffusion enhances the quality of life in surrounding communities by improving building functionality and aesthetics. However, excessive design may raise concerns about cultural heritage and historical continuity, which can affect educational and exhibition functions. Adobe Firefly prioritizes preserving the historical and cultural characteristics of buildings, whereas Mid Journey integrates some modern design elements.

Overall, Stable Diffusion achieves the highest score, followed by Adobe Firefly, with Mid Journey scoring the lowest. Stable Diffusion's rich model and versatile plugins yield ideal results, while Mid Journey's lack of edge detection and structural reference results in significant deviations from actual scenes. Adobe Firefly can generate matching images but lacks the visual richness of the other two tools. In image generation efficiency, Adobe Firefly excels, completing images in about twenty seconds, a significant improvement over traditional design processes, which can take weeks.

## Conclusion

This study explores the application of generative AI in architectural design, using the renewal of industrial architectural heritage in Dalian as a case study [40, 41]. By testing and comparing three mainstream tools—Stable Diffusion, Mid Journey, and Adobe Firefly—and constructing an evaluation system using the Analytic Hierarchy Process (AHP), this paper systematically analyzes the performance, potential, and limitations of these tools in the context of industrial architectural heritage renewal [42, 43], while also offering insights into the contributions and future prospects of this research.

In terms of research contributions, the study first established an evaluation system tailored to industrial architectural heritage renewal [44]. By analyzing national policy documents, highly cited literature, and incorporating expert consultation and field research, four primary evaluation indicators were proposed: architectural renovation, environmental aesthetics, economic benefits, and socio-cultural value. These were further divided into 16 secondary indicators, providing a scientific basis for generating and evaluating design schemes. Analysis of expert questionnaire scores revealed that "architectural aesthetic continuity" and "improvement of quality of life" had the highest weights in the project, with values of 0.380 and 0.363, respectively, indicating that maintaining the historical continuity of architectural aesthetics and enhancing the quality of life for surrounding residents are core demands in the renewal of industrial heritage. In the comparative analysis of generative AI tools, Stable Diffusion scored

the highest in the indicators of "architectural renovation" and "socio-cultural value," with scores of 6.380 and 6.921, respectively, and an overall score of 6.3. Mid Journey showed balanced performance in "environmental aesthetics" and "economic benefits" but scored lower in "architectural renovation," with scores of 5.072 and 4.202. Adobe Firefly excelled in speed and ease of use, making it suitable for quickly generating concept images.

Despite the significant advantages demonstrated by generative AI tools in this study [45, 46], certain limitations remain. First, these tools still struggle with understanding human emotions and cultural contexts, which may lead to design schemes lacking in emotional resonance and cultural symbolism. Second, the randomness and uncontrollability of generated results are challenges that designers must address, especially when using Mid Journey, where multiple iterations and selections are required to achieve the desired outcome. Additionally, the use of generative AI tools places higher demands on designers' skills, requiring them to not only master traditional design theory but also become proficient in leveraging these new tools for scheme optimization. As generative AI technology continues to develop, its application prospects in the renewal of architectural heritage are broad, but further technological optimization and enhanced collaboration between designers and AI are necessary.

Overall, generative AI tools provide new approaches and methods for the renewal of industrial architectural heritage, significantly simplifying the design process while improving efficiency and quality. Although some technical and application limitations still exist, with ongoing advancements, generative AI is poised to play an increasingly significant role in the design field, driving innovation and development in the industry.

## Author Contributions

**Conceptualization:** Pengjun Wu.

**Formal analysis:** Xiaowen Li.

**Funding acquisition:** Wei Mo.

**Methodology:** Yao Liu.

**Project administration:** Wei Mo.

**Software:** Yao Liu.

**Writing – original draft:** Pengjun Wu.

**Writing – review & editing:** Pengjun Wu.

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
