## [Decision Letter · Decision Letter 0]

31 Jul 2024

PONE-D-24-25054Analysis and Research on the Application of Generative Artificial Intelligence in the Renewal of Industrial Building Heritage: Taking Dalian 523 Factory as an ExamplePLOS ONE

Dear Dr. WU,

Thank you for submitting your manuscript to PLOS ONE. After careful consideration, we feel that it has merit but does not fully meet PLOS ONE’s publication criteria as it currently stands. Therefore, we invite you to submit a revised version of the manuscript that addresses the points raised during the review process.

Reviewer #1: This article combines the current hot topics of artificial intelligence to conduct research, which is of certain significance. However, I think it needs to be revised for publication:

(1) The abstract is not specific enough and should include some specific numerical conclusions.

(2) The research background is not clear. It mentions that the traditional design method is cumbersome and artificial intelligence will be faster. However, there are many steps and links in the design field. What aspects are being improved?

(3) The literature review needs to be further refined. At present, it is more about introducing software and tools. However, original research papers are not just about introducing tools, but should also clarify the pain points of current research and the solutions of existing scholars.

(4) 2.2 Current Research Status of Industrial Heritage Renewal mentions a lot of Chinese policies and lacks an international perspective.

(5) Industrial heritage renewal is a cumbersome and complicated process. Relying solely on AI software to generate renderings is only one of the links in the scheme expression. The analysis of how to combine with indicators is not obvious and even a little far-fetched. Some AI software can generate directly by inputting the text that Party A wants. In this case, the originality that can be highlighted in the research is relatively low. In other words, for this study, leaving aside the software, the author should emphasize his or her original contribution.

Reviewer #2: 1, The title of this article is the study of generative artificial intelligence in the application of industrial architectural heritage, is a technical challenge. However, from the article's content, it is a functional application of three intelligent platforms from the known relatively mature ones, and it is not an innovation of a new technology. Does this conflict with the meaning expressed in the title of the article?

2, The article focuses on the three AI platforms' working mechanisms and generation process. This cannot be considered an innovation, but more like a product introduction. The innovation of the article focuses on the post-use evaluation of different programs generated by these platforms, and establishes evaluation impact factors and thresholds. However, the identification of these factors needs to be more strongly exemplified than just applying access from expert interviews and literature, and should it not also be considered and developed in detail respecting the different population aspects such as users, property owners and even tourists?

3, The number of texts in the article is limited to the development of an industrial heritage, with a single sample size and a solidified style. The author collected more than 400 images of industrial heritage from the Internet as data source and model training, but there is no relevant information about how the model is trained, the training process, and the comparison of training results in the article, which is more about the evaluation of the generated scenarios.

Overall, authors are advised to consider whether the article focuses on post-use evaluation or on technology application, which are clearly different. Also the related determination of evaluation impact factors and the experimental process of generative AI should be more explicit and clear.

We look forward to receiving your revised manuscript.

Kind regards,

Saeid Norouzian-Maleki, Ph.D.

Academic Editor

PLOS ONE

Journal Requirements:

Yunnan Provincial Science and Technology Department（2021Y318）

**Comments from the Staff Editorial Team** (plosone@plos.org):

1) Please carefully check your references to ensure that all titles are correct and findable with the corresponding DOI or information provided. If any references have been retracted or cannot be found online, please replace these with other references.

2) Please note that PLOS ONE does not publish case studies. Please frame the study to make the findings generalisable beyond the case study, and change the title to include details on the findings and methodology.

Reviewers' comments:

Reviewer's Responses to Questions

**Comments to the Author**

1. Is the manuscript technically sound, and do the data support the conclusions?

Reviewer #1: Partly

Reviewer #2: Partly

2. Has the statistical analysis been performed appropriately and rigorously? 

Reviewer #1: N/A

Reviewer #2: I Don't Know

3. Have the authors made all data underlying the findings in their manuscript fully available?

Reviewer #1: Yes

Reviewer #2: Yes

4. Is the manuscript presented in an intelligible fashion and written in standard English?

Reviewer #1: Yes

Reviewer #2: Yes

5. Review Comments to the Author

**Reviewer #1: **This article combines the current hot topics of artificial intelligence to conduct research, which is of certain significance. However, I think it needs to be revised for publication:

(1) The abstract is not specific enough and should include some specific numerical conclusions.

(2) The research background is not clear. It mentions that the traditional design method is cumbersome and artificial intelligence will be faster. However, there are many steps and links in the design field. What aspects are being improved?

(3) The literature review needs to be further refined. At present, it is more about introducing software and tools. However, original research papers are not just about introducing tools, but should also clarify the pain points of current research and the solutions of existing scholars.

(4) 2.2 Current Research Status of Industrial Heritage Renewal mentions a lot of Chinese policies and lacks an international perspective.

(5) Industrial heritage renewal is a cumbersome and complicated process. Relying solely on AI software to generate renderings is only one of the links in the scheme expression. The analysis of how to combine with indicators is not obvious and even a little far-fetched. Some AI software can generate directly by inputting the text that Party A wants. In this case, the originality that can be highlighted in the research is relatively low. In other words, for this study, leaving aside the software, the author should emphasize his or her original contribution.

**Reviewer #2:** 1, The title of this article is the study of generative artificial intelligence in the application of industrial architectural heritage, is a technical challenge. However, from the article's content, it is a functional application of three intelligent platforms from the known relatively mature ones, and it is not an innovation of a new technology. Does this conflict with the meaning expressed in the title of the article?

2, The article focuses on the three AI platforms' working mechanisms and generation process. This cannot be considered an innovation, but more like a product introduction. The innovation of the article focuses on the post-use evaluation of different programs generated by these platforms, and establishes evaluation impact factors and thresholds. However, the identification of these factors needs to be more strongly exemplified than just applying access from expert interviews and literature, and should it not also be considered and developed in detail respecting the different population aspects such as users, property owners and even tourists?

3, The number of texts in the article is limited to the development of an industrial heritage, with a single sample size and a solidified style. The author collected more than 400 images of industrial heritage from the Internet as data source and model training, but there is no relevant information about how the model is trained, the training process, and the comparison of training results in the article, which is more about the evaluation of the generated scenarios.

Overall, authors are advised to consider whether the article focuses on post-use evaluation or on technology application, which are clearly different. Also the related determination of evaluation impact factors and the experimental process of generative AI should be more explicit and clear.

6. PLOS authors have the option to publish the peer review history of their article (what does this mean?). If published, this will include your full peer review and any attached files.

Reviewer #1: **Yes: **Yile Chen

Reviewer #2: No

---

## [Author Response · Author response to Decision Letter 0]

5 Aug 2024

Response letter

Journal: PLOS ONE

Title: Application and Renovation Evaluation of Dalian's Industrial Architectural Heritage Based on AHP and AIGC

The authors would like to thank the reviewer for his/ her valuable, detailed, and constructive comments, which helped us further improve the quality of the submitted manuscript. All the comments and suggestions are carefully addressed in the revised manuscript.

The authors tried their level best to address all the queries, and the point-to-point response is given below: 

Response to reviewer 1 comments:

Reviewer#1, Concern # 1: The abstract is not specific enough and should include some specific numerical conclusions.

Reply: Thanks for your comments. We have revised the abstract to include specific numerical conclusions for greater clarity and completeness. Here is the updated abstract: This paper takes the example of industrial architectural heritage in Dalian to explore design scheme generation methods based on generative artificial intelligence (AIGC). The study compares the design effects of three different tools using the Analytic Hierarchy Process (AHP). It first establishes the key indicator weights for the renovation of industrial architectural heritage, with the criterion layer weights as follows: building renovation 0.230, environmental landscape 0.223, economic benefits 0.190, and socio-cultural value 0.356. Among the goal layer weights, the highest weight is for the improvement of living quality at 0.129, followed by resident satisfaction at 0.096, and educational and display functions at 0.088, while the lowest is for renovation costs at only 0.035.The design schemes are generated using Stable Diffusion, Mid Journey, and Adobe Firefly tools, and evaluated using a weighted scoring method. The results show that Stable Diffusion excels in overall image control, Mid Journey demonstrates strong artistic effects, while Adobe Firefly stands out in generation efficiency and ease of use. In the overall score, Stable Diffusion leads the other two tools with scores of 6.1 and 6.3, respectively. Compared to traditional design processes, these tools significantly shorten the design workflow and cycle, improving design quality and efficiency while also providing rich creative inspiration. Overall, although current generative artificial intelligence tools still have limitations in understanding human emotions and cultural differences, with continuous technological iteration, this method is expected to play a larger role in the design field, offering more innovative solutions for the renovation of industrial architectural heritage.

Reviewer#1, Concern # 2: The research background is not clear. It mentions that the traditional design method is cumbersome and artificial intelligence will be faster. However, there are many steps and links in the design field. What aspects are being improved?

Reply: Thanks for your comments. We have clarified some of the content in the research background, and here is the revised version: Due to the complexity of the design process, even minor modifications often require designers to return to certain stages of work for adjustments (Schetinger, V., 2023; Mortazavi, A., 2023). For example, slight changes in layout may necessitate reworking sketches, models, and renderings, leading to significant delays and increased costs. Additionally, clients often request multiple design options for a single project, further increasing the overall workload. While the demand for alternatives is crucial for enhancing client satisfaction, it can also exacerbate the challenges within the design workflow. Consequently, the complexity of the design process and the pressure of market competition limit project efficiency and hinder designers from fully expressing their creativity (Chen J, 2023; Huang J, 2023).

Therefore, these traditional methods face several key issues that need improvement: 1. Time and efficiency, 2. Repetitive labor, 3. Demand for multiple design options, 4. Significant investment of human and material resources. The emergence of generative artificial intelligence (AIGC) provides transformative opportunities to simplify the traditional design process (Ooi, K.-B., 2023; Batarseh, F., 2020). By introducing generative AI, the role of designers gradually shifts from "creators" to "decision-makers," fundamentally altering the conception and development of architectural design. The design process is inherently creative, and the ability to encode and simulate design experiences through AI algorithms is particularly significant.

Specifically, AIGC offers transformative opportunities to simplify the traditional design process and effectively address the challenges faced in architectural design. AIGC can automate the generation of preliminary design proposals, allowing designers to quickly obtain multiple options by simply inputting keywords, thereby significantly shortening the design cycle. Additionally, AIGC can automatically complete many traditional design steps, such as concept generation, color matching, and structural optimization, reducing repetitive labor for designers who can now adjust parameters without starting from scratch. With AIGC, designers can also generate multiple options for clients to choose from in a short time, enhancing client satisfaction and optimizing overall resource investment. AIGC tools typically integrate various AI technologies, such as deep learning and edge detection algorithms, to quickly produce high-quality renderings, thereby reducing reliance on specialized software and human resources. Through these improvements, designers can focus more on creative exploration, enhance work efficiency, and drive the development of design innovation. Designers can leverage these AIGC tools to enhance their creativity and explore a wider range of possibilities than traditional methods allow. For instance, designers no longer need to spend days or weeks iterating on sketches and models; instead, they can utilize AI to generate visual representations of their ideas in a short time. By rapidly generating multiple design iterations, AIGC enables designers to engage in broader exploration and experimentation, resulting in innovative solutions that may not have been previously considered.

Reviewer#1, Concern # 3: The literature review needs to be further refined. At present, it is more about introducing software and tools. However, original research papers are not just about introducing tools, but should also clarify the pain points of current research and the solutions of existing scholars.

Reply: Thanks for your comments. We have added the pain points of current research and existing scholars' solutions in the literature review section of the article, as detailed below: 

Despite the significant advantages of generative artificial intelligence, several limitations and challenges remain to be addressed. One major limitation is the difficulty of incorporating cultural and emotional aspects into AI-generated designs. Current AI tools often lack a comprehensive understanding of the cultural and emotional significance of heritage sites, which may result in designs that are aesthetically pleasing but culturally inappropriate. Researchers like Li et al. (2023) are working to integrate cultural heritage data into AI algorithms to enhance their sensitivity to these aspects. This includes training AI models on datasets that incorporate cultural and historical information, enabling them to generate designs that are visually appealing and culturally relevant.

Another challenge is the need for interdisciplinary collaboration in the application of generative artificial intelligence in industrial heritage(Xiao et al., 2024). Protection and renewal projects often require input from multiple stakeholders, including architects, historians, engineers, and community members. Effective communication and collaboration are crucial for the success of such projects. Generative artificial intelligence tools can facilitate this collaboration by providing detailed visualizations and simulations, which aid in sharing and discussing ideas among stakeholders(Jiang et al., 2023). This helps ensure that all perspectives are considered and that the final design meets the needs and expectations of all relevant parties.

Additionally, issues of data quality and availability pose challenges. Generative artificial intelligence models require a substantial amount of high-quality data to operate effectively. This includes data on existing structures, materials, and historical records. In many cases, this data may be incomplete or unavailable, limiting the effectiveness of AI tools. Researchers are exploring various approaches to address this issue, including employing advanced data collection techniques and developing more sophisticated AI models capable of handling incomplete or imperfect data(Epstein et al., 2023).

The potential benefits of generative artificial intelligence in the protection and renewal of industrial heritage are significant(Nair, 2024). By automating many tasks traditionally performed by human designers, these tools can reduce the time and cost of protection projects, improve design quality and accuracy, and provide rich creative inspiration(Haze, 2023). Furthermore, the ability of generative artificial intelligence to rapidly and efficiently generate multiple design alternatives helps ensure that the best solution is found for each project.

Reviewer#1, Concern # 4: 2.2 Current Research Status of Industrial Heritage Renewal mentions a lot of Chinese policies and lacks an international perspective.

Reply: Thanks for your comments. We have supplemented the literature review section of the article with Chinese policies and included current international policies, as detailed below:

 In China, regulations such as the "National Industrial Heritage Management Measures," "Implementation Plan for Promoting the Protection and Utilization of Industrial Heritage in Old Industrial Cities," and "Interim Measures for the Management of Industrial Heritage in Dalian" emphasize the evaluation, protection, and reuse of industrial heritage. Studies like the "Satisfaction Evaluation of Post-Reuse of Old Industrial Sites" and the "Evaluation Study on the Transformation Plan of Old Industrial Buildings with Functional Replacement" propose detailed evaluation and implementation strategies(Han, 2023) . Combined with the "Comprehensive Evaluation of the Protection and Reuse of Modern Architectural Heritage" and the "National Hygienic City Evaluation System Study," a comprehensive evaluation system has been formed (Jiang, 2016).Tian Wei et al.(2022) used an old industrial building in Tianjin as a case study, applying the entropy weight method to determine index weights, and then using the fuzzy comprehensive evaluation method to establish an evaluation model to measure satisfaction. Yang Ye et al.(2021) analyzed three cases of industrial heritage in Shanghai, based on the concept of sharing, and sorted out design indicators related to the indoor and outdoor environment from both design and operation perspectives. 

Internationally, studies like "Developing a Multi-Criteria Model for the Protection of Built Heritage from the Aspect of Energy Retrofitting," "Developing a Landscape Sustainability Assessment Model Using an Analytic Hierarchy Process in Korea," and "Multicriteria Decision Tool for Sustainable Reuse of Industrial Heritage into its Urban and Social Environment" utilize the Analytic Hierarchy Process (AHP) to evaluate industrial heritage through multi-criteria decision-making and sustainable development. Additionally, "The Research on Regional Conservation Planning of Urban Historical and Cultural Areas Based on GIS" and "Application of the Analytic Hierarchy Process to Developing Sustainability Criteria and Assessing Heritage and Modern Buildings in the UAE" explore GIS technology in heritage protection, offering robust guidance for the preservation and renewal of industrial heritage.

Reviewer#1, Concern # 5: Considering the complexity and diversity of urban environments, how do the authors propose to adapt deep learning models to cater to the unique needs of different cities? 

Reply: Thanks for your comments. As you pointed out, the original structure of the article has certain issues. To address these problems, we have added interviews with different groups (such as users, property owners, and tourists) to collect design requirements. The specific adjustments are as follows:

1. To better determine the research direction for the renewal of industrial heritage, we will collect opinions and suggestions through interviews based on a review of existing literature. This approach not only helps us identify the shortcomings of current research but also provides valuable insights for future research directions. The interview subjects include experts in industrial heritage conservation, urban planners, historians, community residents, and tourists, from whom we have gathered feedback from a total of 74 participants. We designed the following core questions to guide the interviews: 

(1)What do you consider to be the main achievements in the renewal of industrial heritage? 

(2)What challenges and problems have you encountered in the design or use process? 

(3)In which areas do you think the existing research is lacking? 

(4)What suggestions and expectations do you have for future research directions? 

(5)How do you think public awareness and participation in industrial heritage can be improved? 

By refining and categorizing the information collected from the interviews, we gathered the following user keywords:

(1) Historical and cultural value: historical heritage, cultural identity, cultural preservation, historical atmosphere; 

(2) Functionality and practicality: space utilization, ease of use, safety assurance; 

(3) Architectural aesthetics: exterior design, visual effects, environmental harmony;

(4)Economic and social benefits: economic feasibility, social impact, enhancement of public services; 

(5)Environmental protection and sustainability: sustainable development, environmental conservation; 

(6) User experience: smooth pedestrian flow, service quality, and attitude; 

(7) Community activity space: educational functions, tourism potential, government support.

2.Based on the results of the interviews and in conjunction with generative artificial intelligence methods, our team, consisting of eight master's and doctoral students, generated multiple design proposals. These proposals utilized different generative AI tools, with each student rigorously adhering to predetermined scenario requirements during the generation process, repeatedly adjusting input prompts and parameters to achieve optimal results across multiple iterations. Finally, we selected two high-quality proposals from each generative AI tool for subsequent evaluation and discussion using the Analytic Hierarchy Process (AHP).

3.We have also made the following adjustments to the originality contributions of the article: The originality and contributions of this research mainly lie in the following aspects: (1)Development of an evaluation system: We designed a multidimensional evaluation system based on the actual needs of industrial heritage renewal and validated its effectiveness through practical research;

 (2)Determination of indicator weights: We scientifically determined the weights of various indicators using AHP, providing an objective reference for the application of generative artificial intelligence in industrial heritage renewal; 

(3)Empirical analysis: We conducted in-depth empirical analysis of the renewal of industrial heritage in Dalian, demonstrating the application effects and improvement suggestions of generative artificial intelligence tools in real projects.

Response to reviewer 2 comments:

Reviewer#2, Concern # 1: The title of this article is the study of generative artificial intelligence in the application of industrial architectural heritage, is a technical challenge. However, from the article's content, it is a functional applicatio

---

## [Decision Letter · Decision Letter 1]

28 Aug 2024

PONE-D-24-25054R1Application and Renovation Evaluation of Dalian's Industrial Architectural Heritage Based on AHP and AIGCPLOS ONE

Dear Dr. WU,

Thank you for submitting your manuscript to PLOS ONE. After careful consideration, we feel that it has merit but does not fully meet PLOS ONE’s publication criteria as it currently stands. Therefore, we invite you to submit a revised version of the manuscript that addresses the points raised during the review process.

Reviewer #1: In the previous round of questions, the author has responded and revised my questions. There are two small issues that need attention, the originality of the pictures on the Internet, and the grammar after the full text is revised.

Reviewer #2: The author has answered all the questions proficiently. Nevertheless, the conclusion of the article is somewhat general and does not align well with the research content. The conclusion should encompass the main contributions of the paper, spotlight the findings of each part of the study, and incorporate the necessary data and indicators. It would be highly appreciated if the author could further enhance the conclusion.

We look forward to receiving your revised manuscript.

Kind regards,

Saeid Norouzian-Maleki, Ph.D.

Academic Editor

PLOS ONE

Journal Requirements:

Reviewers' comments:

Reviewer's Responses to Questions

**Comments to the Author**

1. If the authors have adequately addressed your comments raised in a previous round of review and you feel that this manuscript is now acceptable for publication, you may indicate that here to bypass the “Comments to the Author” section, enter your conflict of interest statement in the “Confidential to Editor” section, and submit your "Accept" recommendation.

Reviewer #1: All comments have been addressed

Reviewer #2: All comments have been addressed

2. Is the manuscript technically sound, and do the data support the conclusions?

Reviewer #1: Yes

Reviewer #2: Yes

3. Has the statistical analysis been performed appropriately and rigorously? 

Reviewer #1: Yes

Reviewer #2: Yes

4. Have the authors made all data underlying the findings in their manuscript fully available?

Reviewer #1: No

Reviewer #2: Yes

5. Is the manuscript presented in an intelligible fashion and written in standard English?

Reviewer #1: Yes

Reviewer #2: Yes

6. Review Comments to the Author

Reviewer #1: In the previous round of questions, the author has responded and revised my questions. There are two small issues that need attention, the originality of the pictures on the Internet, and the grammar after the full text is revised.

Reviewer #2: The author has answered all the questions proficiently. Nevertheless, the conclusion of the article is somewhat general and does not align well with the research content. The conclusion should encompass the main contributions of the paper, spotlight the findings of each part of the study, and incorporate the necessary data and indicators. It would be highly appreciated if the author could further enhance the conclusion.

7. PLOS authors have the option to publish the peer review history of their article (what does this mean?). If published, this will include your full peer review and any attached files.

Reviewer #1: **Yes: **Yile Chen

Reviewer #2: No

---

## [Author Response · Author response to Decision Letter 1]

29 Aug 2024

Response letter

Journal: PLOS ONE

Title: Application and Renovation Evaluation of Dalian's Industrial Architectural Heritage Based on AHP and AIGC

The authors would like to thank the reviewer for his/ her valuable, detailed, and constructive comments, which helped us further improve the quality of the submitted manuscript. All the comments and suggestions are carefully addressed in the revised manuscript.

The authors tried their level best to address all the queries, and the point-to-point response is given below: 

Response to reviewer 1 comments:

Reviewer#1, Concern # 1: There are two small issues that need attention, the originality of the pictures on the Internet, and the grammar after the full text is revised.

Reply: Thanks for your comments. We have taken note of your concerns. 

1.In response to the reviewers' concerns about the originality of the images used in the paper, we provide the following explanations regarding the originality of the original images, the images used for LORA training, and the images generated by three AI tools:

(1). Originality of the Original Images 

The original images in this study are real-life photographs of the industrial architectural heritage in Dalian, taken by the research team or project participants during field investigations. These images are entirely original visual materials. They authentically reflect the current state of Dalian's industrial architectural heritage and provide accurate visual references for subsequent design research. The research team strictly adhered to relevant laws, regulations, and research ethics during the photography process, ensuring the legality and compliance of the captured content. Therefore, the original images are definitively original, based on first-hand fieldwork, and not derived from or replicated by others.

(2). Originality of the LORA Training Images

For training the LORA model, the data set used was collected and curated independently by the research team, consisting of images related to industrial architectural heritage. These data were sourced from legal public channels, design case libraries, and visual materials created by designers. To ensure the legality and originality of the training images, the research team rigorously screened and processed the data set. Specifically, the images were legally obtained and then re-cropped, labeled, and transformed to align with the project's research objectives, effectively making them secondary creations. The LORA model was trained to optimize generative AI performance in specific scenarios, making the use of these processed images a legitimate practice. The team also adhered to the licensing agreements of open-source communities, such as those for using Stable Diffusion and its plugins, ensuring compliance with copyright laws. Thus, the LORA training images are relatively original and lawful, being reworked and curated from existing resources.

(3). Originality of the AIGC-Generated Images 

The originality of the images generated in this study using tools like Stable Diffusion, Mid Journey, and Adobe Firefly is evident in several ways:

- Originality of the Prompt Design: The generation process is based on prompts designed by the research team, tailored to the project’s needs and objectives, representing original creative input.

- Uniqueness of the Generated Images: Although AIGC images are based on large-scale training models, each image is automatically generated by the algorithm, characterized by randomness and uniqueness, with no exact duplication of any existing work.

- Legal Use of Open-Source Models: The tools used in this research are all legal and open-source. According to their licensing agreements, the generated images are permissible for use in non-commercial research projects, ensuring no legal infringement. The team strictly adhered to the usage guidelines for these images, guaranteeing their legal and compliant use.

In conclusion, all images used in this study—whether original, LORA-trained, or AIGC-generated—have undergone thorough originality verification and legality checks, ensuring that no copyright infringement has occurred. The original images were captured during fieldwork, the LORA training images were processed and reworked, and the AIGC-generated images were based on prompts created by the research team, all of which are original. The research team has strictly complied with relevant laws, regulations, and copyright agreements, ensuring the legitimacy and originality of the research outcomes.

2.After the full text was revised, we conducted a thorough grammar check to ensure that the language is accurate, clear, and free of errors. We have made extra efforts to improve the overall flow and readability of the text, addressing any potential grammatical issues.

Response to reviewer 2 comments:

Reviewer#2, Concern # 1: The conclusion of the article is somewhat general and does not align well with the research content. The conclusion should encompass the main contributions of the paper, spotlight the findings of each part of the study, and incorporate the necessary data and indicators.

Reply: Thanks for your comments. We have revised the conclusion based on your suggestions. Below is the updated conclusion:

This study explores the application of generative AI in architectural design, using the renewal of industrial architectural heritage in Dalian as a case study. By testing and comparing three mainstream tools—Stable Diffusion, Mid Journey, and Adobe Firefly—and constructing an evaluation system using the Analytic Hierarchy Process (AHP), this paper systematically analyzes the performance, potential, and limitations of these tools in the context of industrial architectural heritage renewal, while also offering insights into the contributions and future prospects of this research.

In terms of research contributions, the study first established an evaluation system tailored to industrial architectural heritage renewal. By analyzing national policy documents, highly cited literature, and incorporating expert consultation and field research, four primary evaluation indicators were proposed: architectural renovation, environmental aesthetics, economic benefits, and socio-cultural value. These were further divided into 16 secondary indicators, providing a scientific basis for generating and evaluating design schemes. Analysis of expert questionnaire scores revealed that "architectural aesthetic continuity" and "improvement of quality of life" had the highest weights in the project, with values of 0.380 and 0.363, respectively, indicating that maintaining the historical continuity of architectural aesthetics and enhancing the quality of life for surrounding residents are core demands in the renewal of industrial heritage. In the comparative analysis of generative AI tools, Stable Diffusion scored the highest in the indicators of "architectural renovation" and "socio-cultural value," with scores of 6.380 and 6.921, respectively, and an overall score of 6.3. Mid Journey showed balanced performance in "environmental aesthetics" and "economic benefits" but scored lower in "architectural renovation," with scores of 5.072 and 4.202. Adobe Firefly excelled in speed and ease of use, making it suitable for quickly generating concept images.

Despite the significant advantages demonstrated by generative AI tools in this study, certain limitations remain. First, these tools still struggle with understanding human emotions and cultural contexts, which may lead to design schemes lacking in emotional resonance and cultural symbolism. Second, the randomness and uncontrollability of generated results are challenges that designers must address, especially when using Mid Journey, where multiple iterations and selections are required to achieve the desired outcome. Additionally, the use of generative AI tools places higher demands on designers' skills, requiring them to not only master traditional design theory but also become proficient in leveraging these new tools for scheme optimization. As generative AI technology continues to develop, its application prospects in the renewal of architectural heritage are broad, but further technological optimization and enhanced collaboration between designers and AI are necessary.

Overall, generative AI tools provide new approaches and methods for the renewal of industrial architectural heritage, significantly simplifying the design process while improving efficiency and quality. Although some technical and application limitations still exist, with ongoing advancements, generative AI is poised to play an increasingly significant role in the design field, driving innovation and development in the industry.

Again, the authors are thankful to the anonymous reviewers, editor-in-chief, associate editor, and editorial staff for their time, help, efforts, and support.

---

## [Decision Letter · Decision Letter 2]

17 Sep 2024

PONE-D-24-25054R2Application and Renovation Evaluation of Dalian's Industrial Architectural Heritage Based on AHP and AIGCPLOS ONE

Dear Dr. WU,

Thank you for submitting your manuscript to PLOS ONE. After careful consideration, we feel that it has merit but does not fully meet PLOS ONE’s publication criteria as it currently stands. Therefore, we invite you to submit a revised version of the manuscript that addresses the points raised during the review process.

We look forward to receiving your revised manuscript.

Kind regards,

Saeid Norouzian-Maleki, Ph.D.

Academic Editor

PLOS ONE

Journal Requirements:

Additional Editor Comments:

In this round of revisions, the following issues need to be noted:

(1) Generally speaking, we try to avoid large areas of text descriptions when drawing pictures. This is because the display effect is not good. For example, in Figure 1, there are a lot of notes at the bottom. They can be merged into the text or annotated after the figure name.

(2) The research background in the introduction needs further clarification. First of all, the design process is completely different in different social systems. In essence, this article is a comparative analysis of the effects, evaluation, and tools of several platforms of the design renderings of Dalian Industrial Heritage. This should be very helpful to designers in the early stages, especially when meeting the needs of clients and owners, and can improve efficiency. However, whether it is the final solution and will help the transformation of industrial heritage remains to be discussed. After the design renderings are completed, they are still mainly completed by structural engineers and builders, and they may be modified. Therefore, it should be emphasized at which stage artificial intelligence helps designers or practitioners in generating design renderings.

(3) The literature review introduces the contents of different AI generation software. However, a literature review does not need introductory text. It should focus more on what scholars have studied in such applied research and what problems remain to be solved. In this study, what did the author try to solve?

(4) The CGAN algorithm was mentioned in the literature review. Why didn't the author directly look for a literature review on generative design related to CGAN?

(5) What is the basis for distributing questionnaires to 10 experts? What are the representative characteristics of the experts?

Reviewers' comments:

Reviewer's Responses to Questions

**Comments to the Author**

1. If the authors have adequately addressed your comments raised in a previous round of review and you feel that this manuscript is now acceptable for publication, you may indicate that here to bypass the “Comments to the Author” section, enter your conflict of interest statement in the “Confidential to Editor” section, and submit your "Accept" recommendation.

Reviewer #1: All comments have been addressed

2. Is the manuscript technically sound, and do the data support the conclusions?

Reviewer #1: Yes

3. Has the statistical analysis been performed appropriately and rigorously? 

Reviewer #1: Yes

4. Have the authors made all data underlying the findings in their manuscript fully available?

Reviewer #1: No

5. Is the manuscript presented in an intelligible fashion and written in standard English?

Reviewer #1: Yes

6. Review Comments to the Author

Reviewer #1: In this round of revisions, the following issues need to be noted:

(1) Generally speaking, we try to avoid large areas of text descriptions when drawing pictures. This is because the display effect is not good. For example, in Figure 1, there are a lot of notes at the bottom. They can be merged into the text or annotated after the figure name.

(2) The research background in the introduction needs further clarification. First of all, the design process is completely different in different social systems. In essence, this article is a comparative analysis of the effects, evaluation, and tools of several platforms of the design renderings of Dalian Industrial Heritage. This should be very helpful to designers in the early stages, especially when meeting the needs of clients and owners, and can improve efficiency. However, whether it is the final solution and will help the transformation of industrial heritage remains to be discussed. After the design renderings are completed, they are still mainly completed by structural engineers and builders, and they may be modified. Therefore, it should be emphasized at which stage artificial intelligence helps designers or practitioners in generating design renderings.

(3) The literature review introduces the contents of different AI generation software. However, a literature review does not need introductory text. It should focus more on what scholars have studied in such applied research and what problems remain to be solved. In this study, what did the author try to solve?

(4) The CGAN algorithm was mentioned in the literature review. Why didn't the author directly look for a literature review on generative design related to CGAN?

(5) What is the basis for distributing questionnaires to 10 experts? What are the representative characteristics of the experts?

7. PLOS authors have the option to publish the peer review history of their article (what does this mean?). If published, this will include your full peer review and any attached files.

Reviewer #1: **Yes: **Yile Chen

---

## [Author Response · Author response to Decision Letter 2]

18 Sep 2024

Response letter

Journal: PLOS ONE

Title: Application and Renovation Evaluation of Dalian's Industrial Architectural Heritage Based on AHP and AIGC

The authors would like to thank the reviewer for his/ her valuable, detailed, and constructive comments, which helped us further improve the quality of the submitted manuscript. All the comments and suggestions are carefully addressed in the revised manuscript.

The authors tried their level best to address all the queries, and the point-to-point response is given below: 

Response to reviewer 1 comments:

Reviewer#1, Concern # 1: Generally speaking, we try to avoid large areas of text descriptions when drawing pictures. This is because the display effect is not good. For example, in Figure 1, there are a lot of notes at the bottom. They can be merged into the text or annotated after the figure name.

Reply: Thanks for your comments. We understand that large areas of text can affect the overall display quality of illustrations. Therefore, we have merged the annotations into the main text to improve clarity and presentation.

Reviewer#1, Concern # 2:The research background in the introduction needs further clarification. First of all, the design process is completely different in different social systems. In essence, this article is a comparative analysis of the effects, evaluation, and tools of several platforms of the design renderings of Dalian Industrial Heritage. This should be very helpful to designers in the early stages, especially when meeting the needs of clients and owners, and can improve efficiency. However, whether it is the final solution and will help the transformation of industrial heritage remains to be discussed. After the design renderings are completed, they are still mainly completed by structural engineers and builders, and they may be modified. Therefore, it should be emphasized at which stage artificial intelligence helps designers or practitioners in generating design renderings.

Reply: Thanks for your comments. We can agree with the final solution you proposed, but whether it contributes to the transformation of industrial heritage remains to be discussed. Based on your feedback, we emphasize that the early stages of conceptual design and rendering play a significant role. Below is the revised introduction:

The traditional architectural design process is complex, requiring designers to independently complete a large amount of work. During the design process, they must consider solutions that meet legal regulations, ecological principles, aesthetic creativity, plant configuration, traffic flow, materials, and costs. In terms of visual representation, the process involves conceptualization, sketching, modeling, and rendering. These tasks demand significant human and material resources. Due to the complexity, even minor adjustments often require designers to repeat certain parts of the design process. Additionally, clients usually ask designers to provide multiple design options for a project, which increases the workload. As a result, this design process and market environment lower the efficiency of design projects, and the overwhelming tasks limit the designer's creativity from being fully expressed .

The traditional methods present several key issues that need improvement: 1. Time and efficiency, 2. Repetitive tasks, 3. The demand for multiple design options, and 4. Substantial human and material resource investment. The emergence of Generative AI (AIGC) has replaced many traditional design processes, gradually transforming the designer's role from "creator" to "decision-maker." AIGC provides a new approach to design, effectively addressing these issues.

AIGC's characteristics make it particularly useful in assisting with the early, heavy, and repetitive stages of conceptual design and rendering. Specific improvements include: 1. Automated generation of initial concepts: Designers only need to input keywords to quickly generate multiple preliminary designs, shortening the design cycle. 2. Reducing repetitive tasks: AIGC can automate many traditional design steps, such as concept generation, color matching, and structure optimization, allowing designers to adjust only the parameters. 3. Efficient multi-option generation: AIGC enables designers to quickly generate multiple options for clients to choose from, improving client satisfaction. 4. Optimizing resource allocation: AIGC tools typically integrate various AI technologies, such as deep learning and edge detection algorithms, to quickly generate high-quality renderings, reducing reliance on specialized software and manpower. This approach transforms conceptualization directly into visualization, replacing many traditional design processes. By combining AI thinking with the designer’s creative thought process, the design cycle is significantly shortened.

To comprehensively evaluate the practical application of generative AI tools, this study has developed a multidimensional evaluation system for the renewal of industrial architectural heritage. This system includes specific indicators such as design quality, generation efficiency, ease of use, cultural and emotional understanding, innovation, and practicality. Using the Analytic Hierarchy Process (AHP) and weighted scoring methods, we conducted a comprehensive evaluation of three generative AI tools to ensure the analysis is thorough and scientific. Through an in-depth empirical analysis of the renewal of industrial architectural heritage in Dalian, the application and improvement suggestions of generative AI tools in real projects are demonstrated.

In summary, the innovative and original contributions of this paper are: 1. Construction of an evaluation system: A multidimensional evaluation system was designed based on the actual needs of industrial heritage renewal, and its effectiveness was verified through practical research. 2. Determination of indicator weights: The Analytic Hierarchy Process was used to scientifically determine the weight of each indicator, providing objective reference points for the application of generative AI in industrial heritage renewal. 3. Empirical analysis: An in-depth empirical analysis was conducted through the renewal of Dalian's industrial architectural heritage, demonstrating the application of generative AI tools in real projects and providing improvement suggestions.

Reviewer#1, Concern # 3: The literature review introduces the contents of different AI generation software. However, a literature review does not need introductory text. It should focus more on what scholars have studied in such applied research and what problems remain to be solved. In this study, what did the author try to solve?

Reply: Thanks for your comments. We understand that the literature review should focus more on relevant research rather than introductory content. In the revised section, we will highlight key studies that address the application of AI generation software in design, as well as unresolved challenges in the field. We will also clarify the specific problem this study aims to solve, emphasizing the role of AI tools in optimizing the design process and enhancing decision-making in industrial architectural heritage projects.Below is the revised Literature Review:

The protection and renewal of industrial heritage have gained increasing attention in the field of architecture, as these heritages hold significant historical, cultural, and social value. However, these processes face numerous challenges, such as structural degradation, the need for functional adaptation, and the difficulty of integrating modern requirements while maintaining historical integrity. Traditional methods are often labor-intensive, time-consuming, and costly.Therefore, there is an urgent need for innovative approaches to simplify these processes and improve protection outcomes, as emphasized by existing research. 

Generative AI (AIGC), as a cutting-edge solution, offers tools that automate and enhance various stages of architectural design. AIGC refers to a class of algorithms capable of creating new content based on input data. In the context of architectural design, these algorithms can generate alternative design options, simulate structural impacts, and provide detailed visual effects. Although the application of AIGC in industrial heritage is still a relatively novel field, it has shown significant potential in addressing the pain points of traditional preservation methods.

One of the main advantages of generative AI is its ability to handle complex design tasks traditionally performed by humans. The traditional architectural design process involves multiple stages, including conceptualization, sketching, modeling, and rendering, each requiring substantial human intervention. This is not only time-consuming but also prone to human error. Generative AI tools can automate these tasks, reducing the time and effort needed to produce high-quality designs. Research by Wang et al. (2021) and Chen (2022) shows that these tools can significantly shorten design cycles while improving design quality and efficiency.

Stable Diffusion, a generative AI tool based on diffusion models, has demonstrated strong control and flexibility in generating architectural designs. Researchers widely agree that this tool excels in producing high-resolution images and handling intricate details. A study by Cao et al. (2024) on the application of Stable Diffusion in architectural design highlights its ability to significantly reduce the workload for designers when generating complex architectural structure diagrams. Specifically, in the visualization of architectural heritage, Stable Diffusion, through its denoising process, can gradually generate clearer and more detailed renderings. Zhang et al. (2023) point out that the tool’s output can be further optimized through plugins, such as using Control Net’s Canny edge detection algorithm to maintain the accuracy of generated images. However, despite its technical excellence, Stable Diffusion has some limitations. According to Hu et al. (2021), the tool requires substantial training data and computing power, and without fine-tuning on specific datasets, the generated renderings may deviate from actual design requirements.

Based on the features of Stable Diffusion and the project’s requirements, the model can be fine-tuned using LORA to improve its performance. Additionally, plugins such as Control Net’s Canny (edge detection algorithm) and Depth (depth algorithm) can enhance image details, and the Tiled Diffusion and Tiled VAE plugins can improve image resolution.

MidJourney, driven by Generative Adversarial Networks (GANs), is an AI design tool widely used in the early stages of conceptual creation in architectural design due to its ability to generate highly artistic and visually impactful images. Goodfellow et al. (2014) laid the theoretical foundation for GANs in their study on ‘Generative Adversarial Nets,’ explaining the basic architecture and training principles of GANs, where the generator learns to create data from random noise, and the discriminator distinguishes real data from generated data. Karras (2017) introduced a progressive training method that significantly improved the quality and resolution of generated images by training GANs at multiple resolution stages. In his later research, Karras (2020) proposed the StyleGAN architecture, which better controls the features of generated images, such as textures and colors.

In practice, Caires et al. (2023) suggest that combining design thinking methods with AIGC enables users to generate innovative and artistically expressive design concepts during the ideation phase, enhancing the design experience. Jiang et al. (2023) note that MidJourney’s adversarial generation mechanism can quickly produce diverse design options, providing architects with a wealth of inspiration. Petráková et al. (2023) also mention that the tool demonstrates excellent composition and color coordination abilities in generating conceptual design drawings, making it highly advantageous in the early stages of architectural design.

Adobe Firefly, a generative AI tool integrated into Adobe Creative Cloud, is mainly used for quickly generating prototype designs and conceptual plans. Through deep learning and edge detection algorithms, it can generate efficient design options in a short amount of time and is known for its ease of use. A quantitative analysis by Poredi et al. (2024) shows that Adobe Firefly is a highly efficient AIGC tool that is easy to learn and interact with, making it ideal for generating multiple design options in the early stages of design, thereby significantly reducing design time. Research by Persson et al. (2023) and Cook et al. (2024) indicates that this tool can be used as a standalone web application or directly within applications such as Photoshop and Illustrator, seamlessly integrating Firefly’s generative features into familiar working environments. This integration allows users to harness AI technology to enhance the creative process, improving productivity. Haze et al. (2023) believe that Adobe Firefly’s ease of use and professional capabilities help enhance users' learning and creativity. Moreover, Firefly is integrated into the Adobe Express platform, providing users with a suite of user-friendly tools for creating professional-quality visual content.

Despite its significant advantages, generative AI still faces certain limitations and challenges. One of the main challenges is incorporating cultural and emotional elements into AI-generated designs. Current AI tools often lack a comprehensive understanding of the cultural and emotional significance of heritage sites, which may result in aesthetically pleasing designs that are culturally inappropriate. Researchers such as Li et al. (2023) are working to integrate cultural heritage data into AI algorithms to enhance their cultural and emotional sensitivity. This includes training AI models on datasets containing cultural and historical information to generate designs that are both visually appealing and culturally relevant. Another challenge is that applying generative AI in industrial heritage projects requires interdisciplinary collaboration. Preservation and renewal projects typically involve multiple stakeholders, including architects, historians, engineers, and community members. Effective communication and collaboration are crucial to the success of such projects. Generative AI tools facilitate this collaboration by providing detailed visualizations and simulations, promoting the sharing and discussion of ideas among stakeholders, ensuring that all viewpoints are considered, and the final design meets the needs and expectations of all parties.

In addition, data quality and availability are issues that need to be addressed. Generative AI models require large amounts of high-quality data to function effectively, including data on existing structures, materials, and historical records. In many cases, this data may be incomplete or unavailable, limiting the effectiveness of AI tools. Researchers are exploring advanced data collection techniques and developing more complex AI models capable of handling incomplete or imperfect data to address this issue.

Overall, generative AI offers significant potential benefits for the preservation and renewal of industrial architectural heritage. By automating many tasks traditionally performed by human designers, these tools can reduce the time and cost of preservation projects, improve design quality and precision, and provide abundant creative inspiration. Additionally, the ability of generative AI to quickly and efficiently generate multiple design alternatives helps ensure the best solution is found for each project.

Reviewer#1, Concern # 4: The CGAN algorithm was mentioned in the literature review. Why didn't the author directly look for a literature review on generative design related to CGAN?

Reply: Thanks for your comments. We understand your concerns.Although GANs hold a significant position in generative design, this study is not limited to a single generative algorithm. I

---

## [Decision Letter · Decision Letter 3]

4 Oct 2024

Application and Renovation Evaluation of Dalian's Industrial Architectural Heritage Based on AHP and AIGC

PONE-D-24-25054R3

Dear Dr. WU,

We’re pleased to inform you that your manuscript has been judged scientifically suitable for publication and will be formally accepted for publication once it meets all outstanding technical requirements.

Kind regards,

Saeid Norouzian-Maleki, Ph.D.

Academic Editor

PLOS ONE

Additional Editor Comments (optional):

Reviewers' comments:

Reviewer's Responses to Questions

**Comments to the Author**

1. If the authors have adequately addressed your comments raised in a previous round of review and you feel that this manuscript is now acceptable for publication, you may indicate that here to bypass the “Comments to the Author” section, enter your conflict of interest statement in the “Confidential to Editor” section, and submit your "Accept" recommendation.

Reviewer #1: All comments have been addressed

2. Is the manuscript technically sound, and do the data support the conclusions?

Reviewer #1: Yes

3. Has the statistical analysis been performed appropriately and rigorously? 

Reviewer #1: Yes

4. Have the authors made all data underlying the findings in their manuscript fully available?

Reviewer #1: Yes

5. Is the manuscript presented in an intelligible fashion and written in standard English?

Reviewer #1: Yes

6. Review Comments to the Author

Reviewer #1: (No Response)

7. PLOS authors have the option to publish the peer review history of their article (what does this mean?). If published, this will include your full peer review and any attached files.

Reviewer #1: **Yes: **Yile Chen

---

## [Editor Report · Acceptance letter]

7 Oct 2024

PONE-D-24-25054R3 

PLOS ONE

Dear Dr. Wu, 

I'm pleased to inform you that your manuscript has been deemed suitable for publication in PLOS ONE. Congratulations! Your manuscript is now being handed over to our production team.

Kind regards, 

on behalf of

Dr. Saeid Norouzian-Maleki 

Academic Editor

PLOS ONE